# How Information Framing Nudges Acceptance of China’s Delayed Retirement Policy: A Moderated Mediation Model of Anchoring Effects and Perceived Fairness

**DOI:** 10.3390/bs14010045

**Published:** 2024-01-10

**Authors:** Weixi Zeng, Lixia Zhao, Wenlong Zhao, Yijing Zhang

**Affiliations:** 1Yangtze Delta Region Institute (Huzhou), University of Electronic Science and Technology of China, Huzhou 313001, China; zeng1982@uestc.edu.cn (W.Z.);; 2School of Public Administration, University of Electronic Science and Technology of China, Chengdu 611731, China

**Keywords:** framing effect, policy acceptance, perceived fairness, anchoring effect

## Abstract

China’s delayed retirement policy will be prudently rolled out at the appropriate time, yet the public’s acceptance of this policy is concerning. To address this issue, our endeavor explores the impact of framing and anchoring effects on policy acceptance, aiming to mitigate the populace’s resistance to the new policy. We conducted two survey studies on the Chinese population aged 16–65. Achieved through an online survey, Study 1 (*N* = 225) demonstrated that information framing significantly influences the public’s acceptance of the delayed retirement policy. It was found that perceived fairness plays a mediating role between information framing and policy acceptance. Notably, the positive frame had a more pronounced effect on acceptance than its negative counterpart, with the positive presentation being perceived as more fair. Study 2 (*N* = 383), utilizing a combination of online and offline approaches, revealed that the anchoring effect moderates the relationship between information framing and perceived fairness. The interaction of anchoring and framing effects significantly influences perceived fairness, subsequently promoting public policy acceptance. The interplay between anchoring and framing effects significantly shapes perceived fairness, in turn bolstering the public’s receptiveness to policy. These insights offer reasonable communication strategies for the smooth advancement of new policies, further enriching the field of behavioral science.

## 1. Introduction

Currently, countries around the world face challenges regarding the sustainability of their pension systems, and many nations have begun to promote delayed retirement to alleviate this social issue [1,2]. However, In China, public acceptance of the delayed retirement policy is far from optimistic. As early as a 2013 survey involving ten thousand participants, 94.5% of respondents explicitly expressed opposition to delayed retirement, with nearly 70% opposing flexible retirement. In fact, since 2008, there have been calls for the introduction of a delayed retirement policy, but no formal policy for implementation has been issued at the national level. This indicates China’s extreme caution in formulating its delayed retirement policy. At the first session of the 14th National People’s Congress in 2023, Premier Keqiang Li, when responding to a journalist’s question, stated that the delayed retirement policy would be prudently introduced at an appropriate time. From this, it is evident that the implementation of China’s delayed retirement policy is inevitable, and the policy direction and principles are now clearly established [3].

Delaying the legal retirement age is a common response to population aging and pension pressures in many countries. Yet, in 2023, President Emmanuel Macron’s plan to raise the retirement age faced strong resistance from various sectors of French society, sounding an alarm for other countries about public opinion. The introduction of a new policy will inevitably elicit intense emotional reactions from the public. Hence, navigating the subtle waters of public sentiment to lessen resistance towards policy changes stands paramount for a nation’s stability. As the delayed retirement policy unfolds, it is imperative to delve into research that elevates the public’s receptiveness and fosters a supportive stance.

However, on one hand, the existing research on policy acceptance covers a limited scope and perspective, lacking exploration of individual psychological mechanisms and behavioral willingness from the angle of behavioral public administration. In the literature on retirement policy, the primary focus has been on factors influencing workers’ retirement intentions, designs of delayed retirement schemes, implementation measures, and the feasibility and impacts of delayed retirement [4], with limited research on public support or acceptance for these policies. Additionally, citizens’ willingness to accept a delayed retirement policy has a significant impact on the financial sustainability of the social pension insurance system [5]. On the other hand, studies in social psychology seldom focused on the emotional determinants of policy acceptance at the individual level [6], while the practical implementation of retirement policies necessitates strong societal acceptance as a foundation. What is more crucial is that when formulating and advancing policies, the government’s judicious use of framing and anchoring effects can enhance citizens’ satisfaction and support for these policies [7,8,9].

Therefore, this study primarily addresses three issues: (1) Investigating whether the framing effect, within the realm of public policy, impacts the public’s acceptance of policy. (2) Building on the first question, it further explores the dimension of perceived fairness in the information framing of retirement policies. (3) This research also delves deeper into explaining the mechanism by which the anchoring effect influences information framing and policy acceptance.

The structure of this paper is organized as follows: Section 2 reviews the relevant literature and proposes research hypotheses. Section 3 briefly introduces the methods and designs of the two experiments. Section 4 and Section 5 offer a detailed account of the procedures and results of the two studies. Section 6 and Section 7 discuss and conclude the research results.

## 2. Literature Review and Research Hypothesis

### 2.1. Policy Acceptance

Understanding public acceptance of policies can promote better policy design and communication, secure broader public endorsement, and ensure policy effectiveness [10]. The main findings of these studies can be summarized in the following three aspects:Social psychological elements that sway policy receptiveness. Studies have found that trust and emotions play significant roles in policy acceptance [6,11,12]. Notably, negative emotions exert a more profound direct influence on public policy acceptance than positive ones [6,11].Perceptions of policies and their attributes, such as benefits, costs, effectiveness, and fairness [10]. The perceived fairness of a policy emerges as a paramount influence of its acceptance [10,13,14,15,16]. A thematic analysis was conducted by scholars on over 60 policy articles related to environmental taxes, and it was found that people are more likely to support environmental policies when they perceive the policy to be fair in terms of cost distribution and social sharing [17].Contextual factors of policy acceptance, such as media exposure [18,19] and information framing [20]. Research from Sweden indicated higher public acceptance of a carbon tax policy following the release of the documentary “An Inconvenient Truth” and the publication of “Stern Review” [21]. During the COVID-19 crisis, positive media framing of expert predictions on GDP growth bolstered support for pandemic policies, especially among those expecting an economic downturn [22]. This accentuates the pivotal role of information framing in shaping public opinion and supporting government policies.

In short, previous studies have shown that research on policy acceptance has touched on numerous fields such as food, agriculture, healthcare, and environmental conservation, primarily exploring specific policy issues. However, there is a lack of consideration for the situational conditions of advancing new policies, which may limit the comprehensiveness of understanding policy acceptance. Generally speaking, public opinion is a decisive factor for policy changes in democratic countries; lacking broad public support is a primary obstacle to achieving the objectives of a new policy [23,24].

### 2.2. Attribute Framing Effect and Its Impact on Persuasion

Information framing is a theory-based persuasive communication strategy, aiming to facilitate changes in perception, judgment, attitude, and behavior by presenting equivalent appeals [25]. In the field of behavioral science, framing effects are often used to influence individual decision-making and behavioral preferences [26,27]. For example, by providing individuals with information framed around the benefits and public good nature of vaccines, it is possible to increase residents’ intentions to get vaccinated [28]. Furthermore, framing effects are also widely designed to influence public opinion. For instance, moral foundation framing can not only affect certain groups’ attitudes towards refugees but can also predict the attitudes and behaviors of groups towards the inevitable [29].

Attribute framing is a commonly used type of information framing, focusing on how an individual’s preference might shift when key characteristics of an object or event are described either positively or negatively [30]. The most classic example of the attribute framing effect is the “beef experiment”, in which ground beef samples were rated as tasting better and less greasy when labeled with a positive tag (75% lean) rather than a negative one (25% fat). Another common application of attribute framing involves describing the same scenario based on success rates versus failure rates. For instance, Republicans are more willing to pay additional fees for items labeled as “carbon offsets” rather than “carbon taxes” [25].

Similarly, in the realm of public policy, the prudent application of the attribute framing effect for policy design can enhance public support for policies, thereby guiding decision-making towards the maximization of individual and public welfare. For instance, in policy promotion, portraying the implementation effects of a policy in terms of opportunities is more effective in garnering public support than portraying them in terms of risks [31,32]. Additionally, if policymakers focus on highlighting and communicating the benefits of a policy, rather than seeking to reduce public costs associated with the policy, the public might be more inclined to accept and adhere to it [6].

In specific policy areas, indeed, delayed retirement policies are major initiatives universally adopted by countries around the world. Whether the rationality of the policy can be peacefully accepted by the public, however, requires more astute policy communication. As previously mentioned, policy optimization based on objective facts and policy adjustments made only after the emergence of various social issues represent two different information portrayals and labels. Hence, we argue that presenting the intention behind the delayed retirement policy adjustment using the shortfall in pension funds and aging problems is a portrayal from a negative frame. Framing the policy optimization effort made by the government in light of the objective fact that China’s legal retirement age is misaligned with its labor market structure can cast a more positive light on the delayed retirement policy.

In summary, the following hypotheses are proposed:

**H1.** 
*When an information frame is employed, regardless of its form, the acceptance in the post-test will be significantly higher than the acceptance in the pre-test.*


**H2.** 
*Compared to when receiving a positive information frame, the public has a lower acceptance of the delayed retirement policy when exposed to a negative information frame.*


### 2.3. The Impact of Perceived Fairness on Policy Acceptance and the Framing Effect

Previous research has already revealed the crucial role of perceived fairness in policy acceptance [10,13,14,15,16]. For example, empirical studies on environmental policies have shown that perceived fairness, second only to perceived effectiveness, is a crucial factor directly influencing the acceptance of low-carbon policies [14]. In fact, under the context of climate change taxation and legal frameworks, fairness has become one of the key specific policy predictors of policy acceptance [33]. Additionally, fairness has been considered the strongest predictive factor for accepting policies aimed at promoting healthy food choices [34].

Similarly, in China, researchers investigated residents’ acceptance of Shanghai’s license plate auction policy and then found that policy effectiveness, affordability, fairness concerns, and implementation were factors influencing the acceptance of the vehicle license plate auction policy [35]. More crucially, attribute framing effects also impact individuals’ perceptions of fairness in decisions [36], and constructing the same healthcare resource allocation scenario in positive and negative ways in public health decision-making leads to different perceptions of fairness [37]. Therefore, this study applies information framing to the research on delayed retirement policy and further examines the mediating effect of perceived fairness.

Currently, the proportion of the elderly population in China has reached 14.20%, meeting the standard of a deeply aged society. This demographic trend implies that future workers will have to pay higher taxes to support a growing elderly population. This poses a tremendous challenge to the “pay-as-you-go” social pension insurance system, potentially leading to intergenerational inequity [38]. Given China’s initial aim to establish a basic old-age insurance system founded on social mutual assistance and intergenerational fairness [5], the perception of fairness in the delayed retirement policy plays a crucial role in public support. The new policy mandates delayed retirement, raising two key fairness concerns among the public. Firstly, intergenerational fairness: younger workers may feel their career progression is hindered, while older workers might feel compelled to work despite poor health. Secondly, the personal impact of delayed retirement, including sacrifices in life planning, increased work intensity, and health challenges, could amplify public grievances about fairness.

Thus, we propose the following hypotheses:

**H3.** 
*Individuals exposed to the negative framing information will perceive lower policy fairness compared to those exposed to positive framing information.*


**H4.** 
*Perceived fairness mediates the relationship between information framing and policy acceptance. That is, regardless of the type of attribute framing, it will influence the acceptance of the delayed retirement policy through perceived fairness.*


### 2.4. The Anchoring Effect and Its Impact on Persuasion

Research in cognitive psychology reveals that an individual’s decision-making is influenced by the manner in which information is presented, encompassing two primary cognitive biases: framing bias and anchoring bias [39]. The anchoring effect stands as one of the most potent cognitive heuristics and is ubiquitously present in human decision-making processes [40]. When provided with a reference point, individuals, intentionally or unintentionally, tend to gravitate towards this anchor, consequently affecting their initial cognitive attitudes.

Research has shown that significant interactions between framing effects and anchoring effects can lead people to have different attitudes and responses to different presentations of information. When information framing describes product attributes in a positive or negative manner and is consistent with anchoring information, they jointly impact internet consumers’ attitudes, purchase intentions, and willingness to pay [39]. However, other scholars have shown that while framing and anchoring effects can influence consumers’ attitudes and intentions towards purchasing organic foods, their interactive effect is not significant [41]. Based on current research, we can infer that both the framing effect and the anchoring effect, as cognitive biases, will influence public acceptance.

Globally, in terms of delayed retirement policies, the standard retirement age in developed countries during the same period is generally set at 65 years old. Moreover, many countries are extending the standard retirement age to 67 years. For instance, our neighboring country, Vietnam, which is also a socialist country, amended its labor law as early as November 2019 to implement a delayed retirement policy. Starting in 2021, the retirement age for Vietnamese men was 60 years and three months, and it will be delayed by three months each year, reaching 62 years by 2028. Similarly, the retirement age for women in 2021 was 55 years and four months, delaying retirement by four months each year until reaching 60 years in 2035. In comparison, China’s statutory retirement age remains at 60 for male workers, 50 for female workers, and 55 for female cadres, which is noticeably low.

It is beneficial to incorporate an international perspective into the frames rather than just focusing on domestic views. By framing the information within an international perspective rather than limiting it to the domestic context, the higher retirement ages abroad can serve as an anchor point to influence the public’s perception of delaying retirement, thereby enhancing the perceived fairness among the public. By altering the public’s mental reference point for retirement age, it becomes more conducive to evaluate their distribution fairness in relation to others during policy adjustments [42]. In other words, we believe that promoting the global trend of delayed retirement, rather than just focusing on domestic perspectives, is more beneficial for the public to accept the postponement of the statutory retirement age.

As such, we propose the following hypotheses:

**H5.** 
*The anchoring effect moderates the relationship between framing types and the perception of fairness.*


**H6.** 
*When presented with a lower retirement age anchor, both positive and negative frames will result in a heightened inclination towards policy acceptance.*


## 3. Research Methodology

### 3.1. Research Design

This study employed a survey experimental method and gathered data using both online and offline methods. In terms of research subjects, the study referenced the demographic group from the 2022 China Retirement Readiness Index Survey Report [43], covering individuals aged 16–65, including those who are unemployed. Considering the information receptivity and reading capabilities of middle-aged and elderly people under the current retirement system in China, as well as the lower-than-expected age of delayed retirement among the youth [43], we reduced the sampling proportion of the 56–65 age group and increased the survey emphasis on younger workers. Operationally, data collection in the survey experiment was executed through the distribution of questionnaires. Unlike traditional surveys, survey experiments allow researchers to manipulate variables, creating distinct versions of the questionnaire, thereby clearly discerning the effects of different levels of independent variables on dependent variables [44]. Compared to laboratory studies, survey experiments face greater challenges in terms of scenario initiation. Therefore, this research draws from real-world policy suggestions and presents experimental scenarios in textual form to ensure the “authenticity” of the materials and the participants’ sense of immersion.

### 3.2. Research Overview

To investigate and analyze the influence and mechanism of the framing and anchoring effects on the public’s acceptance of the delayed retirement policy, we conducted two experiments. In Study 1, we primarily investigated whether a positive/negative frame would significantly affect the Chinese public’s acceptance of the delayed retirement policy (H2) and whether any type of information framing would enhance policy acceptance (H1). Additionally, we validated the perceived fairness under different frames (H3) and the mediating role of perceived fairness between framing effects and policy acceptance (H4). In Study 2, we examined the moderating role of the anchoring effect on two types of frames (positive/negative) and perceived fairness (H5) and tested the willingness of policy acceptance for different framing types when there was an anchor (H6). The conceptual hypothesis model of this study is shown in Figure 1. We recruited participants from both offline and online platforms in the two experiments and employed the statistical analysis software SPSS 27.0.1 to analyze the results.

## 4. Study 1

The purpose of Study 1 was to preliminarily verify that the framing effect could significantly influence people’s acceptance of policies. In addition, we sought to explore the impact of framing effects on perceptions of fairness and the mediating role of perceived fairness between framing effects and policy acceptance.

### 4.1. Participants

A one-factor between-subjects design (a positive frame vs. a negative frame) was used in this experiment. Using the online platform Credamo, we distributed a total of 268 questionnaires and received all 268 back. After excluding 43 invalid questionnaires due to incorrect answers, implausible extreme values, and missing data, a total of 225 valid responses remained (*N*_positive_ = 112, *N*_negative_ = 113), yielding an effective response rate of 83.95%. Out of the 225 participants, 125 were male, accounting for 55.6% of the sample. Moreover, 28.4% of the participants were aged between 26–35 years, 28.9% held managerial positions, and most participants rated their health condition as relatively good (M = 3.43, SD = 0.93). Demographic details of the participants in Study 2 are presented in Table 1. Participants received red envelope rewards as a token of appreciation for their participation in the study.

### 4.2. Procedure and Measures

Given that individuals’ reactions to the same scenario can differ based on the manner in which information is presented, these frames guide distinct policy discussions and policy support. Accordingly, this experiment designed two stimulus materials, corresponding to two experimental conditions: the problem description framing, or the negative frame text, which describes the policy intent of delaying retirement in terms of social issues like pension deficits and aging population; and the fact presentation framing, or the positive frame text, primarily emphasizing the government’s policy optimization based on changing social realities. To ensure the authenticity and readability of the materials, they were all derived from official documents or expert interpretations. Complete experimental materials for Study 1 can be found in Table A1.

A one-factor between-subject design was adopted in Study 1, with the participants being randomly assigned to either the “fact presentation” positive frame or the “problem description” negative frame condition. We primarily utilized the Credamo data platform for the online survey experiments. Credamo is a professional data collection platform in China, akin to Amazon Mechanical Turk. Through its “random block” feature, Credamo can randomly allocate participants to different experimental groups, facilitating the randomization of our experiment. The sample quality from this platform has received recognition from many international journals in the social science domain, including Public Management [45]. The experimental procedure first presented the participants with instructions and introduced the policy background. Next, it gauged the participants’ personal and family circumstances as well as their acceptance of the delayed retirement policy (termed pre-acceptance). The principal questionnaire items were rated on a 7-point scale.

Pre-acceptance evaluated the current willingness to accept the delayed retirement policy from three aspects: attitude for the policy, support for the policy, and willingness to respond to the policy’s call (Cronbach’s α = 0.96). Perceived fairness was assessed using three items: “Do you believe the ‘progressive delayed retirement policy’ presented in the material is fair?”, “Do you think the ‘progressive delayed retirement policy’ in the material aligns with the public interest of our nation?”, and “Do you believe the ‘progressive delayed retirement policy’ in the material can promote overall societal fairness?” (Cronbach’s α = 0.93).

Subsequently, participants were exposed to the experimental material. After reading, they were asked to answer two framing effect manipulation check items. The first question was, “Which description do you think best matches the ‘progressive delayed retirement policy’ mentioned in the material above?” The options were: “1 = ‘proposed based on changing social realities for policy optimization’, 2 = ‘proposed to address social issues like pension deficits and aging population’, and 3 = ‘none of the above’”. The second question was, “From which perspective do you think the ‘progressive delayed retirement policy’ in the above material is described?” Participants were asked to rate on a scale from 1 (negative) to 7 (positive).

Finally, after reading the materials, participants were queried on their willingness to accept the retirement policy presented (termed post-acceptance), measured in correspondence with the pre-acceptance items (Cronbach’s α = 0.95). To reduce the range restriction in attitude change and explain the variance in post-test policy acceptance, we incorporated pre-acceptance as a control variable in the model, with post-acceptance as the dependent variable [46].

### 4.3. Result

#### 4.3.1. Manipulation Check

A one-way analysis of variance (ANOVA) was conducted using the SPSS 27.0.1 software. The results of the ANOVA showed that under different types of information frames, the participants’ perceptions of the positive frame were significantly higher than that of the negative frame (*M*
_positive_ = 5.33, *M*
_negative_ = 1.83, F(1, 223) = 481.90, *p* < 0.001, $\eta_{p}^{2}$ = 0.68). This indicates that the manipulation of frame type was effective.

#### 4.3.2. The Main Effect

The ANOVA results revealed significant differences in policy acceptance under the positive and negative frames (*M*
_positive_ = 3.89, *M*
_negative_ = 3.15, F(1, 223) = 38.50, *p* < 0.001, $\eta_{p}^{2}$ = 0.15, as shown in Figure 2). Specifically, the positive frame enhanced citizens’ acceptance of the retirement policy, providing further support for H2. We also found that post-acceptance was significantly higher than pre-acceptance only in the positive frame condition (t _positive_ = 8.20, *p* < 0.001; t _negative_ = 0.05, *p* > 0.05, see Table 2). Thus, H1 received partial support. Additionally, the participants believed that policy communication under a positive frame was perceived as fairer than under a negative frame (*M*
_positive_ = 3.87, SD = 1.52; *M*
_negative_ = 3.06, SD = 1.45, t = 4.10, F(1, 223) = 16.78, *p* < 0.001, $\eta_{p}^{2}$ = 0.07, as illustrated in Figure 2). The results of these data validated H3.

Next, we employed Model 4 of the PROCESS macro in SPSS 27.0.1 to test the mediating effect of perceived fairness. We chose a bootstrap sample size of 5000 and set the confidence interval to 95%. The research results indicated, as shown in Table 3 and Figure 3, that the direct impact of the framing effect on policy acceptance (β = 0.34, SE = 0.09, LLCI = 0.16, ULCI = 0.53) was highly significant.

More importantly, the indirect impact of the framing effect on policy acceptance through perceived fairness was also significant (β = 0.39, SE = 0.09, LLCI = 0.22, ULCI = 0.59). These findings provide support for H4.

## 5. Study 2

The impact of different information frames on policy acceptance was confirmed by Study 1 (H1&H2), and the framing effect on perceived fairness was also evaluated (H3). Furthermore, it assessed the mediating role of perceived fairness between frame effects and policy acceptance (H4). Study 2 aimed to explore and validate the moderating role of the anchoring effect on the relationship between the framing effect and perceived fairness (H5) and to test whether, with a lower retirement age anchor, both frames would lead to a higher willingness to accept the policy (H6). The study adopted a 2 (positive frame vs. negative frame) × 2 (with anchor vs. without anchor) between-subjects design. Participants were randomly assigned to one of the four combination scenarios.

### 5.1. Participants

Study 2 was conducted both online and offline. Online data collection was conducted through the Credamo platform, accumulating 256 responses, while offline data were gathered at the Chengdu People’s Park and the entrance of Chengdu East Station subway from random pedestrians across the country, collecting a total of 216 responses. After excluding the questionnaires with wrong answers to the screening questions, unreasonably extreme values, and missing values, 89 were discarded (41 from online and 48 from offline). This left us with a total of 383 valid responses (*N*_positive × without anchoring_ = 95, *N*_negative × without anchoring_ = 95, *N*_positive × with anchoring_ = 97, *N*_negative × with anchoring_ = 96). The effective recovery rate for the online survey was 83.98%, and for offline it was 77.78%. Out of the 383 respondents, 52.0% were male. Additionally, 30.0% of the participants were aged between 26–35 years, and 33.2% held non-managerial positions. Furthermore, the majority rated their health condition as relatively good (*M* = 3.45, SD = 0.92). The demographic details of the respondents from Study 2 can be seen in Table 1.

### 5.2. Procedure and Measures

The experimental method and procedure for Study 2 were the same as Study 1. The difference in the experimental materials lay in adding a section of text information before the framing effect scenario material (as shown in Table A2), serving as the main manipulation material for the moderating variable of the anchoring effect. From an international perspective, the retirement age and policy in Vietnam were taken as an anchor point, emphasizing the policy differences in the retirement age between countries of socialist governance. From a domestic perspective, no comparison was made between China and other countries’ retirement age policies.

In Study 2, an anchoring effect manipulation check was added: “Which of the following descriptions do you think best matches the retirement age in China as depicted in the material above?” with options being: 1 = The current statutory retirement age in China is 60 for men and 50 for female workers. 2 = The current statutory retirement age in China is lower than that of Vietnam. 3 = The current statutory retirement age in China is higher than that of Vietnam. Other items were set up similarly to Study 1, and their details can be found in Table A1. In Study 2, the internal consistency, as measured by Cronbach’s α, was commendable for the pre-acceptance (α = 0.96), perceived fairness (α = 0.93), and the post-acceptance (α = 0.95).

### 5.3. Result

#### 5.3.1. Manipulation Check

The effectiveness of the information framing manipulation was assessed by comparing scores from the positive- and negative-framed test questions. A one-way ANOVA analysis revealed significant differences in the participants’ perceptions of whether the stimulus material used a positive or negative frame (*M*_positive_ = 5.51, *M*_negative_ = 2.10, F(1, 381) = 624.94, *p* < 0.001, $\eta_{p}^{2}$ = 0.62), indicating a significant validity of the framing manipulation.

#### 5.3.2. The Main Effect

We conducted a moderated mediation analysis using Model 7 in PROCESS, employing 5000 bootstrap samples and a 95% confidence interval, to test the full model. The independent variable was the type of framing effect, the dependent variable was the post-acceptance, the moderator was the anchoring effect, and the control variable was the participants’ pre-acceptance. As shown in Table 4, perceived fairness significantly increased as the frame became more positive (β = 0.73, *p* < 0.001). As expected, the anchoring effect moderated the impact of different framing types on perceived fairness (β = 0.66, *p* < 0.001), thereby supporting H5. Moreover, we found a statistically significant relationship between frame type × anchor type and perceived fairness (β = 0.55, *p* < 0.05). Once perceived fairness was incorporated into the regression model, policy acceptance significantly rose as the framing effect transitioned from negative to positive (β = 0.48, *p* < 0.001). To capture the interaction effect better, a spotlight analysis was performed, the results of which are presented below (see Figure 4). Without a retirement age anchor, there were significant differences in the individual impacts of the positive and negative frames on perceived fairness (β = 0.73, SE = 0.17, LLCI = 0.39, ULCI = 1.06); with an age anchor, there were also significant differences in perceived fairness across the different frame types (β = 1.27, SE = 0.17, LLCI = 0.93, ULCI = 1.61, *p* < 0.001).

Additionally, in the absence of an anchor, the indirect effect of the framing effect on retirement policy acceptance through perceived fairness was 0.35. With an anchor, the indirect effect was 0.61. Both effects were significant, thus confirming H6. The index of the moderating mediation effect was 0.26 at a significance level of 0.05, indicating that the anchoring effect significantly moderated the indirect effect of the information framing on attitudes toward delayed retirement policy via perceived fairness (β = 0.26, SE = 0.12, LLCI = 0.02, ULCI = 0.50). As illustrated in Figure 5, regardless of the presence or absence of an anchor, a positive frame led to higher policy acceptance intention. However, when an anchor was present, it significantly boosted citizens’ acceptance of delayed retirement across different framing types.

## 6. Discussion

From a behavioral science perspective, this study innovatively explores how framing effects and perceived fairness collectively influence individual policy acceptance intentions and probes the dimensions of the anchoring effect within. Firstly, our results provide solid evidence for the significant impact of framing effects on policy acceptance. Previous studies have shown that information framing can determine the focus of public attention, altering attitudes or intentions by selectively emphasizing certain aspects or details [47,48]. In determining which strategies are more effective, we found that the positive frame promoted citizens’ acceptance intentions more effectively than the negative one, aligning with findings from some existing studies [27,37,49,50]. Upon exposure to a positive frame, the public’s policy acceptance significantly increased compared to the initial levels, whereas a negative frame did not substantially alter their acceptance intentions. This might be attributed to the worrisome and sensitive nature of the information presented in negative frames, magnifying issues like governmental inaction and flaws in institutional design. Negative labels might evoke more adverse and detrimental associations [25,37]. Thus, policymakers (and their advisors) should prudently recognize the potential varied impacts of policy labels on different groups; seemingly trivial semantic differences to one individual might have profound effects on another [49]. When an object or event is presented in a positive way, it evokes more favorable associations in the participants’ memories, rather than negative ones. Therefore, effective policy communication is crucial for fostering the target audience’s understanding of the policy.

Moreover, fairness is a central moral issue in public policies [51], acting as a pivotal factor promoting policy support [10,35], and it is also a crucial piece of information that can be presented in the attribute frame. This paper argues about how positive and negative frames regarding the delayed retirement policy influence the public’s perceived fairness, consequently altering their policy acceptance. The research identified that under the “fact-presentation” positive frame, the public’s perceived fairness of the policy exceeded that under the “problem-description” negative frame. Therefore, in the context of the delayed retirement policy, employing a policy communication strategy that uses a positive ‘fact presentation’ frame is more effective. In the negative frame, people tended to attribute the emergence of the delayed retirement policy to the government’s responsibility, overlooking the public issue that arises from social development and requires efforts from multiple parties to solve. Once negative labels such as “government inaction” or “government management failure” are attached to the delayed retirement policy, it can lead to public doubts about the government’s governance capabilities and protests against the perceived unfairness of the policy. Previous research has already revealed the important role of perceived fairness in policy acceptance [10,13,14,15,16]. However, on the one hand, previous studies have primarily focused separately on the fields of psychology (framing effects/anchoring effects) and public policy (policy acceptance), lacking an effective mechanism that centers on individual cognitive biases, perceived fairness, and policy acceptance from the perspective of behavioral public administration. On the other hand, although attribute framing has been tested in many other fields [25], research on attribute framing’s effect on perceived policy fairness remains limited.

Lastly, our study found that presenting lower retirement ages from other countries can effectively moderate the framing effect and enhance the public’s perception of fairness. By shifting the public’s mental reference point regarding retirement age, it is more conducive for the public to perceive distributive fairness during the policy adjustment process [42]. However, research has shown that providing policy information from developed countries significantly reduced the public’s judgment on the appropriateness of property taxes for China’s context [52]. Therefore, it is essential to be aware of the heterogeneous impacts of an international perspective on public policy support during policy communication and promotion.

In conclusion, this study, from a behavioral science perspective, sought to enhance public acceptance of new policies by proposing the impact of information framing and anchoring effects based on citizens’ behavioral preferences and cognitive biases. We emphasize that different information frames can promote public’s policy acceptance and that providing informational anchors is beneficial for policy support. Moreover, the importance of perceived fairness in reducing public resistance should be given significant attention. These findings are valuable in supplementing and enriching policy communication, offering more evidence for the application of behavioral public policy in gaining policy support. Additionally, while this study focused on the delayed retirement policy, the insights and conclusions gained may offer valuable perspectives and approaches for understanding and promoting other types of new policies.

## 7. Conclusions

### 7.1. Implications

In past research, scholars have deeply probed the relationship between the framing effect and policy acceptance. Additionally, connections between the framing effect and perceived fairness, as well as between perceived fairness and policy acceptance, have garnered attention. Yet, despite these invaluable insights, there remains a conspicuous research lacuna concerning the intricate interplay among the framing effect, perceived fairness, and policy acceptance. Thus, a seminal contribution of this paper lies in its intricate exploration of how the framing effect and perceived fairness jointly influence individual policy acceptance intentions whilst also shedding light on the noteworthy role of the anchoring effect. This provides both theoretical and practical guidance for policy communication and behavioral public administration.

By harnessing the information framing, combined with perceived fairness and the anchoring effect, we delved into the mechanisms driving public policy acceptance, synthesizing the “fact-presentation” in policy framing with the “international perspective” in policy propagation. Both theoretically and practically, this study broadens the horizons of the existing research on policy support and framing effects, aptly reflecting the recent trend in behavioral public policy research and practice to merge macro (i.e., retirement policy) with micro perspectives (i.e., personal willingness to accept the policy) [47,53].

Moreover, the findings of this paper can provide effective communication strategies for the smooth implementation of delayed retirement policies and have the potential to be applied to other new policy implementation scenarios. This paper unveils the pivotal role of fairness perception in policy support, a connection that deserves close attention in future policy communication and dissemination efforts. Additionally, it emphasizes that, in policy communication, a judicious focus on specific policy facets or details, approached from the citizen’s vantage point and interests, paves the way for heightened policy endorsement and implementation. In sum, when the government crafts policy, adeptly leveraging framing effects and informational anchor points can markedly amplify citizens’ satisfaction and backing for policies [7,52].

### 7.2. Limitations and Future Research

This study has several limitations. First, to adapt to the actual situation, we increased the proportion of younger participants in the survey. Although we ensured that there were no differences in the experimental samples across groups, this might have impacted the representativeness of the results. Since this study focused only on the Chinese population aged from 16 to 65, this limits the applicability of the results in other cultural and demographic contexts. In future research, more representative or diverse samples and different methods should be used to validate and expand these findings. Secondly, the dependent variable in this study was the participants’ self-reported willingness to accept the policy, but it remains uncertain whether this truly represents their actual behavior. While willingness is often considered a precursor to behavior, there is a difference between behavioral intentions and actual behavior. It is worth noting that in this study, we focused on the public’s willingness to accept new policies, not their actual behavior. Fortunately, we found that using low-cost, contextually adaptable behavioral science methods and nudging strategies can significantly garner more public support. In this study, we conducted both pre-tests and post-tests on participants’ willingness, utilizing their pre-acceptance as a control variable, thereby enhancing the validity of our experiments. Additionally, the experimental materials were meticulously designed based on genuine public information and combined with operational requirements. Although we did not conduct actual interventions, this research is reliable given the current background of China’s upcoming push for delayed retirement policies. Nonetheless, we cannot ascertain whether participants’ processing of these materials deviates from their reactions to intervention information in real-world scenarios. Furthermore, experimental stimuli in survey experiments typically show rapid effects but are short-lived, which to some extent limits the impact that can be drawn from the results. Thus, future studies should resort to field experiments, allowing participants to receive intervention information in real-world situations, making the experimental stimuli more authentic and enduring.

## Figures and Tables

**Figure 1 behavsci-14-00045-f001:**
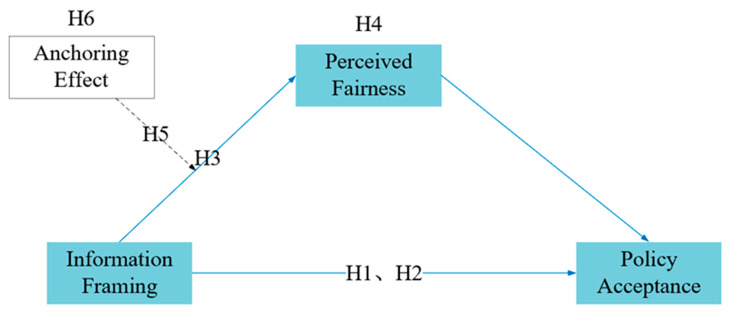
The hypothetical theoretical model.

**Figure 2 behavsci-14-00045-f002:**
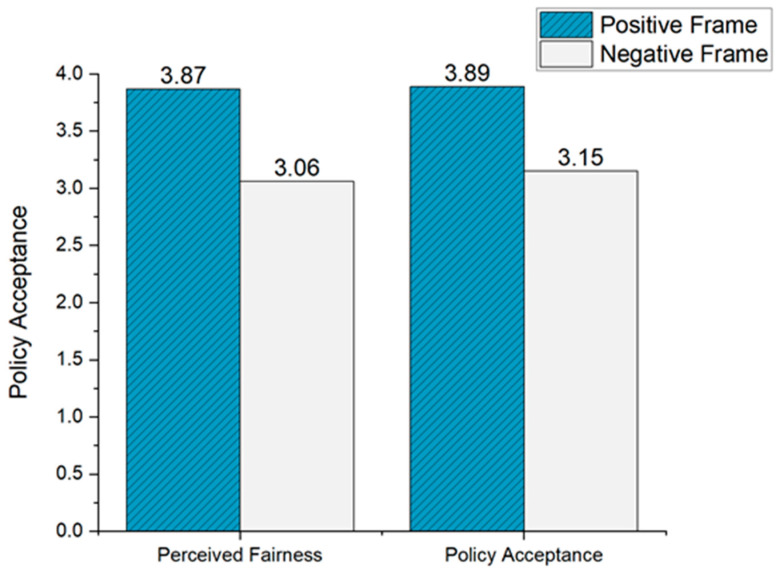
Policy acceptance and perceived fairness before and after exposure to different information framing (Study 2).

**Figure 3 behavsci-14-00045-f003:**
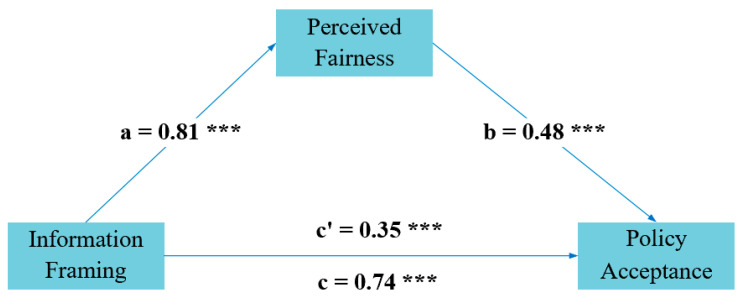
Regression analysis of the mediating effect. Notes: *** *p* < 0.001.

**Figure 4 behavsci-14-00045-f004:**
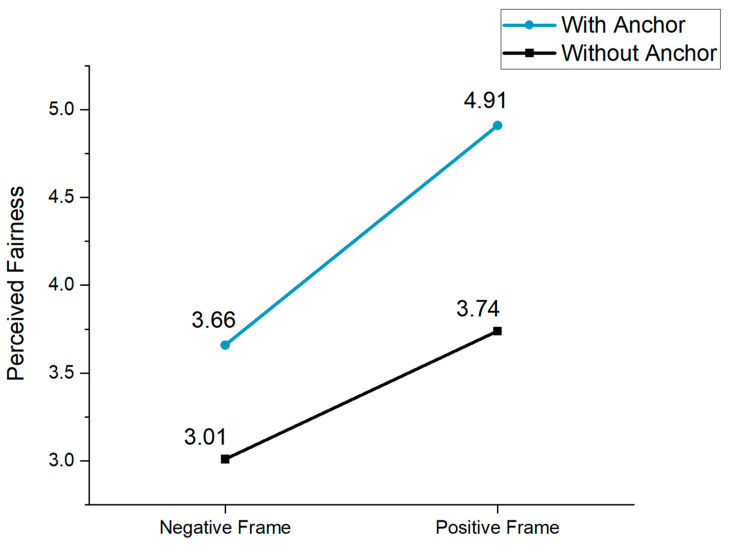
The effects of framing and anchoring on perceived fairness (Study 2).

**Figure 5 behavsci-14-00045-f005:**
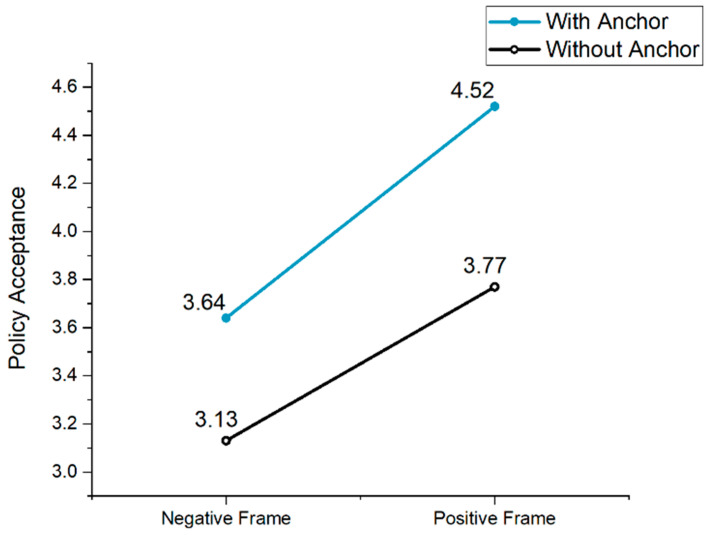
The effects of framing and anchoring on policy acceptance (Study 2).

**Table 1 behavsci-14-00045-t001:** Demographic information of subjects (Study 1, Study 2).

Variables	Levels	Study 1 (*N* = 225)		Study 2 (*N* = 383)	
		*N*	*%*	*N*	*%*
Gender	Male	125	55.6%	199	52.0%
	Female	100	44.4%	184	48.0%
Age	16–25	54	24.0%	80	20.9%
	26–35	64	28.4%	115	30.0%
	36–45	46	20.4%	86	22.5%
	46–55	44	19.6%	78	20.4%
	56–65	17	7.6%	24	6.3%
Education level	Junior high School or below	33	14.7%	54	14.1%
	High school/vocational secondary school	52	23.1%	76	19.8%
	Bachelor’s degree/ Associate degree	113	50.2%	199	52.0%
	Master’s degree or above	27	12.0%	54	14.1%
Job position	Unemployed	32	14.2%	59	15.4%
	Management	65	28.9%	127	33.2%
	Non-management	128	56.9%	197	51.4%

**Table 2 behavsci-14-00045-t002:** Policy acceptance before and after exposure to different information framing (Study 2).

Experimental Scenario	Policy Acceptance	M ± SD	t	*p*
Positive frame (N = 112)	Pre-acceptance	3.14 ± 2.04	8.20	<0.001
	Post-acceptance	3.89 ± 1.97		
Negative frame (N = 113)	Pre-acceptance	3.15 ± 1.85	0.05	0.96
	Post-acceptance	3.15 ± 1.72		

**Table 3 behavsci-14-00045-t003:** Regression analysis of the mediating effect.

Effect (Framing→Fairness→Acceptance)	B	SE	LLCI	ULCI
Direct effect of X on Y	0.35	0.09	0.17	0.53
Indirect effect of X on Y	0.39	0.09	0.23	0.59
Total effect of X on Y	0.74	0.12	0.51	0.98

**Table 4 behavsci-14-00045-t004:** Results of regression analysis (Study 2).

	Perceived Fairness		Post-Acceptance	
	β(SE)	t	β(SE)	t
Framing	0.73 (0.17) ***	4.22	0.31 (0.08) ***	3.76
Perceived fairness			0.48 (0.03) ***	16.06
Anchoring	0.66 (0.17) ***	3.84		
Framing × anchoring	0.55 (0.24) *	2.24		
Pre-acceptance	0.34 (0.03) ***	10.98	0.62 (0.02) ***	27.94
R2	0.3958		0.8365	
F	61.90 ***		646.22 ***	

Notes: * *p* < 0.05, *** *p* < 0.001.

## Data Availability

Data supporting the reported results are available from the authors on request.

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
