# Peer review of "How Information Framing Nudges Acceptance of China’s Delayed Retirement Policy: A Moderated Mediation Model of Anchoring Effects and Perceived Fairness"

_behavsci, 2024, doi:10.3390/bs14010045_

Round 1

Reviewer 1 Report

Comments and Suggestions for Authors

Dear authors,

I read your article about the important topic of delaying the retirement age in China for both men and women, a problem that is common to other countries as well. I believe that your paper would be improved by mentioning that other countries like Chile and France and other countries in the OECD are also facing stress to their retirement system due to the political difficulties to extend the retirement age.

The main questions is whether some forms of policy communication improve the perception of fairness for extending the retirement age in China.

The topic is relevant, because policy makers (not only in China, but also in France, Latin America and other countries) have political difficulties to extend the retirement age.

The work reports on the results of three online experiments using information framing, perceived fairness, and a mix of both. The methodology is adequate, but the survey datasets are limited, so it is not possible to add much further controls. The conclusions are adequate to the main data results.
The references are appropriate. I suggested some additional literature.   It would be better for the authors to compare the sample of the three online surveys with the demographics of China using the latest Census data, since it is likely that the survey samples are biased towards more educated and wealthy people. This non-representativeness of the survey samples for the Chinese population can bias the results and weakens the policy recommendations from this article.

Author Response

Dear Reviewer,

Please accept our apologies for any confusion caused by the line numbering issues in the previously submitted Word document due to revision marking. We kindly request that you refer to the attached latest version of our coverletter(author-coverletter-33059468.v3). Thank you!

Reviewer 2 Report

Comments and Suggestions for Authors

This article focuses on the delayed retirement policy in China and the concern regarding public acceptance of this policy. To address this issue, three survey studies were conducted on the Chinese population aged 16 to 65. In the first study, it was found that the way information is presented significantly influences the public's acceptance of the delayed retirement policy. The second study demonstrated that the perception of fairness plays a mediating role between information framing and policy acceptance. Notably, the positive framing had a more pronounced effect on acceptance than its negative counterpart and was perceived as fairer. The third study revealed that the anchoring effect moderates the relationship between information framing and the perception of fairness. The interaction of anchoring and framing effects significantly influences the perception of fairness, subsequently promoting public policy acceptance. In summary, these studies suggest that the way information is presented and the anchoring effect can influence the perception of equity of a policy, which, in turn, affects public acceptance of that policy. These findings offer reasonable communication strategies to facilitate the implementation of new policies and enrich the field of Behavioral Science.

Strengths of the Article:  

-Focus on a relevant topic: The article addresses a current and relevant topic, which is the delayed retirement policy in China, making it pertinent to the current discussion on public policies and retirement.

-Strong research methodology: The study employs a robust research methodology, including three survey studies, which provide a broad and diverse database to analyze the relationship between information presentation, perception of equity, and policy acceptance.

-Significant findings: The study's findings are interesting and shed light on how the way information is presented and the anchoring effect can influence public acceptance of a policy, which can be valuable for policymakers.  

Weaknesses of the Article:  

-Geographic limitation: The study focuses solely on the Chinese population aged 16 to 65, which limits the applicability of the results to other cultural and demographic contexts.

-Lack of specific details: The provided summary is somewhat general and lacks specific details about the methodologies used, precise results, or practical implications of the findings.

-Absence of mention of limitations: The summary does not mention potential study limitations, such as sample biases or other methodological considerations that could affect the validity of the results.  

It would be beneficial to improve the following aspects of the article in line with the previous comments:  

-Enhanced Contextualization: Providing a brief introduction that describes the broader context of the delayed retirement policy in China and why it is important would be useful. This would help readers understand the relevance of the study from the outset.

-Reinforced Bibliography: Ensuring that the complete article includes references to relevant prior research in the fields of behavioral science and public policy can enrich the context and reader's understanding. It is essential to delve into a comprehensive and up-to-date bibliography on the topic in question.

-Practical Implications: The article should include a discussion of the practical implications of the findings for policymakers and other stakeholders. How can these results be used to improve the acceptance of the delayed retirement policy in China or in other contexts?

-Discussion of Limitations: It is essential that the article discusses the study's limitations, such as potential sample biases, methodological constraints, or any other factors that could affect the validity of the results. This would contribute to a more balanced assessment of the findings.

The research proposes objectives and results in accordance with the scientific method. The study carries out an investigation respecting the structure and methodological criteria described in the first sections. The proposed results are relevant and presented correctly. It is important to do a minor review of the bibliography, adding new, more current references from impact journals. It would also be good to improve the conclusions and make a comparison with other related theories. In general terms, the article is good, but it would be good to delve into the two aspects mentioned that need improvement

Author Response

Dear Reviewer,

Please accept our apologies for any confusion caused by the line numbering issues in the previously submitted Word document due to revision marking. We kindly request that you refer to the attached latest version of our coverletter. Thank you!

Reviewer 3 Report

Comments and Suggestions for Authors

I had the honor to review the manuscript “How Information Framing Nudges Acceptance of China's Delayed Retirement Policy: A Moderated Mediation Model of An-choring Effects and Perceived Fairness”. The paper addresses an important topic – attitudes towards the delaying retirement in China.

Overall, I did like the paper. The literature is extensive, the methods sound and the results well presented. The text is well written. I only have a few minor comments on the transferability of the results:

·       My first comment is already mentioned in the limitations section. It is questionable if (how) well the results of the attitudes are transferable to actual behavior. This is mentioned in the limitations sections, but only briefly and I would like to see a few sentences more here.

·       Furthermore, I would like the authors to discuss the duration of the effects found in the experiments. I would argue that these are rather short them limiting the implications that are drawn from the results.   

Comments on the Quality of English Language

Some minor typos were found

Author Response

(The authors gave the same response as above.)
